# Research on Performance of Cooperative FSO Communication System Based on Hierarchical Modulation and Physical Layer Network Code

**DOI:** 10.3390/s22186912

**Published:** 2022-09-13

**Authors:** Huaijun Qin, Yang Cao, Xiaofeng Peng, Zupeng Zhang

**Affiliations:** 1School of Electrical and Electronic Engineering, Chongqing University of Technology, Chongqing 400054, China; 2Periodical Agency of Chongqing University of Technology, Chongqing University of Technology, Chongqing 400054, China

**Keywords:** cooperative FSO communication system, data priority, hierarchical modulation, physical-layer network code, strong turbulence

## Abstract

To solve the problem that the channel conditions in asymmetric cooperative FSO communication systems are not fully utilized, and the data reliability deteriorates due to high-order modulation, we proposed a layered modulation, joint physical-layer network coding scheme. In this scheme, we first designate the data priority of the information to be transmitted at the source node. Then, the transmission power of different proportions is allocated to the data based on its priority. Then, the modulated data is sent to each node, and physical-layer network coding is performed on the received data at the relay node. Finally, the relay node sends the encoded information to the destination node, and the destination node recovers the original information using the physical-layer network coding scheme. The simulation results showed that when the average signal-to-noise ratio of the channel was 15 dB, the BER of the cooperative FSO communication system could be reduced to below $10^{-8}$. In the strong atmospheric turbulence channel, the cooperative FSO communication system can obtain a signal-to-noise ratio gain of about 1.5 dB. Under strong atmospheric turbulence, this scheme could also improve the average channel capacity performance of a cooperative FSO communication system.

## 1. Introduction

As a line-of-sight transmission technology, free space optical (FSO) communication technology has the advantages of no spectrum authorization, good confidentiality, and flexible networking. This technology uses a laser as a transmission carrier, and the optical signal is susceptible to atmospheric turbulence when transmitted in free space [1,2,3]. The irregular random movement of atmospheric turbulence will lead to random fluctuations in the optical signal, which will cause problems such as changes in the arrival angle of the beam and fluctuations in the light intensity at the receiving end, thus forming an atmospheric turbulence effect. The atmospheric turbulence effect will give rise to the bit error rate of the entire optical communication system, affecting the quality of communication [4,5]. To suppress the effect of atmospheric turbulence on the FSO communication system, many researchers have applied the collaborative communication method to the FSO communication system. In a collaborative FSO communication system, the relay node tends to be closer to the source node than the destination node. Due to the change in communication distance, the atmospheric turbulence affected by the laser transmission in the atmosphere is also different. The longer the communication distance, the stronger the atmospheric turbulence effect. Therefore, in this case, the channel loss of the direct-pass link and the relay transmission link will be different, which will lead to asymmetric reception problems at the destination node.

To solve the problem of communication system performance degradation caused by asymmetry of transmission channels, hierarchical modulation technology can be applied to collaborative communication systems. In reference [6], Hua Sun and Soon Xin Ng proposed a collaborative communication system based on single-relay auxiliary hierarchical modulation. By using turbo code and mesh code modulation (TTCM) schemes at the source nodes, using the decode-and-forward collaboration protocol in the relay node, the power consumption of the entire system is reduced by 3.62 dB per time slot. In reference [7], Hua Sun and Soon Xin Ng proposed a three-layer collaborative communication system assisted by dual relays; the system integrates the three schemes of TTCM three-layer HM-64QAM and two-layer SPM-16QAM, with a rate of 0.5 under the collaborative communication mode. Finally, the system is optimized to reliably transmit layer three HM-64QAM signals using two-time slots, with an average signal-to-noise ratio of 6.94 dB per slot. In reference [8], Md.Jahangir Hossain and Mohamed-Slim Alouini proposed a scheme that uses adaptive hierarchical modulation to simultaneously transmit voice and multiple types of data on fading channels by changing the constellation size and the priority parameters of the hierarchical signal constellation; the scheme gives the interrupt probability of speech and data transmission on the Nakagami-m fading channel, the achievable spectral efficiency, and the average bit error rate (BER) closed expression and numerical results. In reference [9], Ahmet Zahid and Melda investigated the diversity gain maximization problem in wireless relay systems and used a scheme of setting thresholds to mitigate error propagation in hierarchical modulation, and its analysis and simulation results show that the threshold must depend on the hierarchy constant.

In the actual communication system, the asymmetry of the channel quality will lead to the deterioration of the performance of the communication system. Although the use of hierarchical modulation technology can effectively suppress the deterioration of asymmetric channel performance, it does not solve the problem of system throughput deterioration. Based on this, many researchers have applied network coding technology combined with hierarchical modulation technology to collaborative communication networks. In reference [10], Jung Min Park and Seong-Lyun Kim combined layered modulation with network code to achieve the spectral efficiency of the bidirectional relay channel through network coding and solve the asymmetry of the bidirectional relay channel through layered modulation technology. Finally, the end-to-end error probability and spectral efficiency of the asymmetric relay channel are significantly improved. In reference [11], Tang M and Chen J H studied the proposed joint hierarchical modulation and physical-layer network coding (HM-PNC) scheme under the asymmetric additive Gaussian white noise (AWGN) bidirectional relay channel; it not only ensures the efficient transmission of the better-quality channel but also ensures the reliable transmission of the poor-quality channel. In reference [12], Selvakumar Tharranetharan and Md. Jahangir designed a bit interleaved coded modulation (HMNC-BICM) scheme based on hierarchical modulation and network coding assistance. For coded modulation TWRC systems, this scheme has better performance in terms of average bit error rate. In reference [13], Tang Meng designed a physical-layer network coding (2/4PSK-PNC) scheme using 2/4PSK layered modulation under asymmetric channels, derived the BER performance of the relay and end-to-end under AWGN channels and Rayleigh fading channels, and used it to solve the performance degradation of the communication system.

The above document applies the hierarchical modulation network coding scheme to the two-way relay channel; although the performance of the communication system is improved, there is still some room for improvement in its performance. Moreover, the hierarchical modulation, combined with the network coding scheme, is not applied to the free space optical communication system, and the performance improvement needs to be further studied. This paper, aiming at the asymmetric reception problem at the destination node, proposes a hierarchical modulation joint physical-layer network coding scheme to improve system performance. The scheme allocates different transmission power according to the relative importance of the information to be sent to ensure the high reliability of the data. This paper applies the scheme to the collaborative FSO communication system. The hierarchical modulation mode is used at the source node and the relay node, and the physical-layer network encoding method is used for the received information at the relay node. Finally, the original information is recovered at the destination node. According to the 4/16-QAM modulation method used in this scheme, the bit error rate and average channel capacity performance of the collaborative FSO communication system using the hierarchical modulation-combined physical-layer network coding scheme are analyzed. However, under the conditions of different signal transmission power, different turbulence intensity, and different channel average signal-to-noise ratios, the priority parameters of different constellations and the asymmetric parameters of channels are systematically simulated, and the simulation results show that this scheme is suitable for strong atmospheric turbulence channels and can solve the problem of bit error rate performance degradation caused by the asymmetry of the channel in collaborative FSO communication systems. It can also improve the average channel capacity of the system.

## 2. System Model

The relay network model based on collaborative FSO communication used in this article is shown in Figure 1, where the four source nodes $S_{1}$, $S_{2}$, $S_{3}$, and $S_{4}$, with the assistance of the relay node, exchange information with each other. Information is exchanged with each other with the assistance of relay nodes, and all nodes use the half-duplex working mode between them. FSO communication is used between all nodes, and the two source nodes on the diagonal need to exchange information through relay nodes and neighboring source nodes. Due to the increase in atmospheric turbulence intensity caused by the increase in communication distance, the channel condition of the relay channel in the system is better than the channel condition of the direct-pass link between the two adjacent user nodes, which leads to the asymmetric problem of the system. The asymmetric problem will degrade the bit error rate performance of the system, assuming that there is an asymmetric problem between the source nodes $S_{1}$, $S_{2}$ and the relay node R. If the collaboration system formed between $S_{1}$, $S_{2}$, and R uses low-order modulation, then the good channel conditions from $S_{1}$ to R and R to $S_{2}$ will not be fully utilized, resulting in a waste of channel conditions; if the high-order modulation method is used, the direct-pass link between $S_{1}$ and $S_{2}$ cannot withstand this high-order modulation method, which will lead to the reliability of the receiving data not being guaranteed, resulting in a decrease in system performance.

In the Figure 1 system, FSO communication was used between all nodes, and the communication link obeyed the Gamma-Gamma distribution model. The light intensity distribution of the GG channel model was related to the size scale vortex, the irradiance of the size scale of the light intensity was in line with the gamma distribution, and the probability density functions of the light intensity received by the GG channel model were as follows [14]:(1)$$f\left(h\right)=\frac{2{\left(\alpha \beta \right)}^{\left(\alpha +\beta \right)/2}}{\Gamma \left(\alpha \right)\Gamma \left(\beta \right)}h^{\left(\frac{\alpha +\beta }{2}\right)-1}K_{\alpha -\beta }\left(2\sqrt{\alpha \beta h}\right)$$

In Equation (1), h is the channel state information of the communication link, $K_{\alpha -\beta }$ represents the modified second-class Bezier function, and Γ(⋅) is the gamma function. α and β represent the effective number of large- and small-scale vortices, respectively, and can be represented by Equations (2) and (3) below:(2)$$\alpha ={\left[\exp\left(\frac{0.49\sigma_{l}^{2}}{{\left(1+1.11\sigma_{l}^{12/5}\right)}^{7/6}}-1\right)\right]}^{-1}$$
(3)$$\beta ={\left[\exp\left(\frac{0.51\sigma_{l}^{2}}{{\left(1+0.69\sigma_{l}^{12/5}\right)}^{5/6}}-1\right)\right]}^{-1}$$

In the above equation, the Rytov variance is represented and can be defined as:(4)$$\sigma_{l}^{2}=1.23C_{n}^{2}k^{7/6}L^{11/6}$$
where $C_{n}^{2}$ is the atmospheric refractive index structure constant, k=2π/λ is the number of light waves, λ is the laser wavelength, and L is the transmission distance of the link.

## 3. Hierarchical Modulation and Physical Layer Network Code Scheme

To solve the asymmetric problem in the collaborative FSO communication system, considering the channel characteristics of the FSO communication system, the hierarchical modulation-combined physical-layer network coding scheme was used in the cooperative FSO communication system to achieve the improvement of the performance of the cooperative FSO communication system. In the system shown in Figure 1, hierarchical modulation was used at the source nodes $S_{1}$, $S_{2}$, $S_{3}$, and $S_{4}$, and hierarchical modulation was used at the relay node in a manner that combined the physical-layer network encoding. To further suppress the interference of atmospheric turbulence on the FSO communication system and obtain higher power utilization, the hierarchical modulation process used 4/16-QAM modulation. The hierarchical modulation and physical layer network code (HM-PNC: hierarchical modulation and physical layer network code) scheme was implemented as follows: Each source node divided the information to be sent into two levels based on the relative importance of the data, namely high-priority data ($S_{ih}$) and low-priority data ($S_{il}$), where i∈(1,2,3,4). Each source node contained 4 bits of information, assuming that the first two bits were high-priority data and the last two bits were low-priority information. Both $S_{ih}$ and $S_{il}$ used 4QAM modulation. We assumed that the transmit power distribution factor between high and low priorities was α, and the final message sent at the source node was:(5)$$S_{i}=\sqrt{\alpha }S_{ih}+\sqrt{1-\alpha }S_{il}$$

The specific principle of 4/16-QAM hierarchical modulation is shown in Figure 2. Figure 2a is a constellation diagram of the high priority data $S_{ih}$, and the arrow refers to the mapping point of data 00 after 4QAM modulation; Figure 2b is a constellation diagram of low priority data $S_{il}$, and the arrow refers to the mapping point of data 01 after 4-QAM modulation; Figure 2c is the constellation chart of the data to be sent $S_{i}$, which is consistent with the constellation chart modulated by 16QAM. In Figure 2, $2d_{2}$ represents the minimum distance between constellation points of high-priority data, with $d_{1}=\sqrt{\alpha /2}$; $2d_{2}$ represents the minimum distance between constellation points of low-priority data within the same quadrant, with $d_{2}=\sqrt{\left(1-\alpha \right)/2}$ [15]; and $2d_{2}$ represents the minimum distance between constellation points for low-priority data in adjacent quadrants.

In a collaborative FSO communication system, since the channel conditions between the relay node and each source node are better, each source node in Figure 1 can obtain all transmission information, including high priority and low priority data. In the system shown in Figure 1, the four source nodes in the first stage send information to the relay node; the second stage relay node sends information to the four source nodes. Each source node can use a direct-through link to obtain high-priority information for neighbor nodes, so only the low-priority information of neighbors needs to be obtained through the relay node; however, the complete information of the remote node must be obtained through the relay node. In the Figure 1 system, the encoding codeword encoded by the physical-layer network at the relay node has the following structure ($\xi_{i}$):(6)$$\xi_{1}=S_{1h}⊕S_{3l}$$
(7)$$\xi_{2}=S_{1l}⊕S_{3h}$$
(8)$$\xi_{3}=S_{2h}⊕S_{4l}$$
(9)$$\xi_{4}=S_{2l}⊕S_{4h}$$

According to the hierarchical modulation scheme used in this system, the relay node can convert four network-encoded codewords into two mixed messages ($J_{i}$), and the specific implementation process is as follows:(10)$$J_{i}=\sqrt{\alpha }\phi_{i}+\sqrt{1-\alpha }\varphi_{i},\;\;\;i=1,2$$

Thereinto, $\phi ,\varphi \in \xi =\left\{\xi_{1},\xi_{2},\xi_{3},\xi_{4}\right\}$ and $\phi_{1}\neq \varphi_{1}\neq \phi_{2}\neq \varphi_{2}$. Let us assume that $\phi_{1}=\xi_{1},\varphi_{1}=\xi_{2},\phi_{2}=\xi_{3},\varphi_{2}=\xi_{4}$ can get $J_{1}=\sqrt{\alpha }\xi_{1}+\sqrt{1-\alpha }\xi_{2}$ and $J_{2}=\sqrt{\alpha }\xi_{3}+\sqrt{1-\alpha }\xi_{4}$.

After HM-PNC, the relay node transmits the encoded information to the four user nodes. The received information for each user node can be expressed as:(11)$$\begin{array}{cc} y_{RN}\left(i\right) & =h_{RN}J_{i}+n_{N}\\ & =h_{RN}\left(\sqrt{\beta }\phi_{i}+\sqrt{1-\beta }\varphi_{i}\right)+n_{N},\;\;\;i=1,2 \end{array}$$
where $h_{RN}$ is the channel fading coefficient from the relay node R to the user node $S_{N}$, N∈{1,2,3,4}. $n_{N}$ is the Gaussian white noise corresponding to the channel.

Finally, each user node can be decoded using serial interference cancellation (SIC), in the following steps:

The high-priority information required by the user node is decoded as:(12)$$\hat{\phi_{i}}=\arg\min{\left\|y_{RN}-h_{RN}\sqrt{\alpha }\phi_{i}\right\|}^{2}$$

Each user node uses the SIC method to eliminate high-priority information from $y_{RN}\left(i\right)$, and the low-priority information required by the user is:(13)$$\hat{\varphi_{i}}=\arg\min{\left\|y_{RN}-h_{RN}\left(\sqrt{\alpha }\hat{\phi_{i}}+\sqrt{1-\alpha }\varphi_{i}\right)\right\|}^{2}$$

Each user node can decode data from other nodes using high-priority data obtained by a direct link with neighboring nodes. For example, the user node $S_{1}$ has four messages, $S_{1h}$, $S_{1l}$, $S_{2h}$ and $S_{4h}$, from neighboring nodes. By decoding $S_{2l}⊕S_{4h}$⨁ $S_{4l}⊕S_{2h}$⨁ $S_{1l}⊕S_{3h}$, and $S_{1h}⊕S_{3l}$, you can get $S_{2l}$⨁ $S_{4l}$⨁ $S_{3l}$, and $S_{3h}$. The details are as follows:(14a)$$S_{2l}=S_{4h}⊕\left(S_{2l}⊕S_{4h}\right)$$
(14b)$$S_{4l}=S_{2h}⊕\left(S_{4l}⊕S_{2h}\right)$$
(14c)$$S_{3l}=S_{1l}⊕\left(S_{1l}⊕S_{3h}\right)$$
(14d)$$S_{3h}=S_{1h}⊕\left(S_{1h}⊕S_{3l}\right)$$

Other user nodes require the same information as the user node $S_{1}$.

## 4. System Performance Analysis

### 4.1. System Bit Error Rate Analysis

In an FSO communication system, the average bit error rate of the received signal can be expressed in Equation (15) [16]:(15)$$\bar{P_{e}}=\int_{0}^{\infty }BER\left(SNR\right)f\left(h\right)dh$$

In this formula, BER(SNR) is the unconditional bit error rate when the channel signal-to-noise ratio is SNR, and f(h) is the probability density function of the channel state information h. The instantaneous signal-to-noise ratio of a channel can be defined as $\gamma ={\left(hI\right)}^{2}/N_{0}$, and the average signal-to-noise ratio can be defined as $\bar{\gamma }={\left(E\left[h\right]I\right)}^{2}/N_{0}$, where I is the light intensity of the emission symbol “1”, $N_{0}$ is the variance of the channel noise, and E[⋅] is the expectation of h. In the case of fully correlated atmospheric turbulence fading, considering only the case of E[h]=1 [17],$h=\sqrt{\gamma /\bar{\gamma }}$ can be obtained, so Equation (1) can be written as:(16)$$f\left(\gamma \right)=\frac{2{\left(\alpha \beta \right)}^{\left(\alpha +\beta \right)/2}}{\Gamma \left(\alpha \right)\Gamma \left(\beta \right)}\frac{\gamma^{\frac{\alpha +\beta }{4}-1}}{{\bar{\gamma }}^{\frac{\alpha +\beta }{4}}}K_{\alpha -\beta }\left(2\sqrt{\alpha \beta \sqrt{\frac{\gamma }{\bar{\gamma }}}}\right)$$

Substitute Equation (16) into Equation (15) to obtain the bit error rate expression of the FSO communication system in 4/16-QAM modulation mode:(17)$${\bar{P}}_{e}=\frac{{\left(\alpha \beta \right)}^{\frac{\alpha +\beta }{2}}}{4\sqrt{\pi }\Gamma \left(\alpha \right)\Gamma \left(\beta \right)}\int_{0}^{\infty }BER\left(\sqrt{SNR}\right)\frac{\gamma^{\frac{\alpha +\beta }{4}-1}}{{\bar{\gamma }}^{\frac{\alpha +\beta }{4}}}G\begin{array}{c} 2,0\\ 0,2 \end{array}\left[\alpha \beta \sqrt{\frac{\gamma }{\bar{\gamma }}}|\begin{array}{c} -\\ \frac{\alpha -\beta }{2},-\frac{\alpha -\beta }{2} \end{array}\right]d\gamma$$

In Equation (17), according to document [18], it is possible to obtain a 4/16-QAM modulation of the bit error rate expression for the constellation priority parameter λ as follows:(18)$$BER\left(SNR\right)=erfc\left(\sqrt{G\left(a,b;\lambda ,M\right)\bar{\gamma }}\right)$$
erfc(⋅) is the complementary error function, where
(19)$$G\left(a,b;\lambda ,M\right)=\frac{{\left(a+b\lambda \right)}^{2}}{2{\left[1+\lambda \left[\frac{\sqrt{M}}{2}-1\right]\right]}^{2}+\frac{2}{3}\left[\frac{M}{4}-1\right]\lambda^{2}}$$

In the above equation, M is the modulation order, a is a non-negative integer, b is a positive integer, and the constellation priority parameter $\lambda =d_{2}/d_{1}$. Substituting Equation (18) into Equation (17), the FSO communication system in 4/16-QAM hierarchical modulation under the bit error rate expression is:(20)$$\begin{array}{l} {\bar{P}}_{e}\left(a,b;\lambda ,M,\gamma \right)=\\ \frac{{\left(\alpha \beta \right)}^{\frac{\alpha +\beta }{2}}}{4\sqrt{\pi }\Gamma \left(\alpha \right)\Gamma \left(\beta \right)}\int_{0}^{\infty }erfc\left(\sqrt{G\left(a,b;\lambda ,M\right)\bar{\gamma }}\right)\frac{\gamma^{\frac{\alpha +\beta }{4}-1}}{{\bar{\gamma }}^{\frac{\alpha +\beta }{4}}}G\begin{array}{c} 2,0\\ 0,2 \end{array}\left[\alpha \beta \sqrt{\frac{\gamma }{\bar{\gamma }}}|\begin{array}{c} -\\ \frac{\alpha -\beta }{2},-\frac{\alpha -\beta }{2} \end{array}\right]d\gamma \end{array}$$

In the system shown in Figure 1, the constellation diagram of each source node $S_{i}$ using a 4/16-QAM hierarchical modulation of the noninverting subflow is shown in Figure 3,

According to the document [18], the bit error rate of high-priority data from source node $S_{i}$ to relay node R can be expressed as:(21)$$P\left\{\varepsilon_{S_{i}R}^{S_{ih}}|\lambda \right\}=\frac{1}{4}\left({\bar{P}}_{e}\left(a_{1},b_{1};\lambda ,M,\bar{\gamma }\right)+{\bar{P}}_{e}\left(a_{2},b_{2};\lambda ,M,\bar{\gamma }\right)\right)$$

Thereinto, $a_{1}=1$, $b_{1}=2$ and $a_{2}=1$, $b_{2}=0$. The bit error rate for low-priority data from source node $S_{i}$ to relay node R can be expressed as:(22)$$P\left\{\varepsilon_{S_{i}R}^{S_{il}}|\lambda \right\}=\frac{1}{2}{\bar{P}}_{e}\left(a_{1},b_{1};\lambda ,M,\bar{\gamma }\right)-\frac{1}{4}{\bar{P}}_{e}\left(a_{2},b_{2};\lambda ,M,\bar{\gamma }\right)+\frac{1}{4}{\bar{P}}_{e}\left(a_{3},b_{3};\lambda ,M,\bar{\gamma }\right)$$

Thereinto $a_{1}=0$, $b_{1}=1$; $a_{2}=2$, $b_{2}=3$; and $a_{3}=2$, $b_{3}=1$.

We can set $\mu ={\bar{\gamma }}_{ij}/{\bar{\gamma }}_{S_{i}R}$ to the channel asymmetric parameter between the adjacent source nodes $S_{i}$ through $S_{j}$ and the source node $S_{i}$ to the relay node R link, thereinto i,j∈{1,2,3,4}. By bringing in Equations (21) and (22) for parameters a and b and introducing asymmetric parameters of the channel, you can get high and low priority data about the bit error rate expressions for asymmetric parameters:(23)$$P\left\{\varepsilon_{S_{i}R}^{S_{ih}}|\lambda ,\mu \right\}=\frac{1}{4}\left({\bar{P}}_{e}\left(1,0;\lambda ,M,\mu {\bar{\gamma }}_{S_{i}R}\right)+{\bar{P}}_{e}\left(1,2;\lambda ,M,\mu {\bar{\gamma }}_{S_{i}R}\right)\right)$$
(24)$$P\left\{\varepsilon_{S_{i}R}^{S_{il}}|\lambda ,\mu \right\}=\frac{1}{2}{\bar{P}}_{e}\left(0,1;\lambda ,M,\mu {\bar{\gamma }}_{S_{i}R}\right)-\frac{1}{4}{\bar{P}}_{e}\left(2,3;\lambda ,M,\mu {\bar{\gamma }}_{S_{i}R}\right)+\frac{1}{4}{\bar{P}}_{e}\left(2,1;\lambda ,M,\mu {\bar{\gamma }}_{S_{i}R}\right)$$
where ${\bar{\gamma }}_{S_{i}R}$ is the average signal-to-noise ratio from source node $S_{i}$ to relay node R.

The total bit error rate of a single source node can be defined as the probability that the node will not be able to successfully receive information transmitted by other nodes. Taking source node $S_{1}$ as an example, the bit error rate expression from other source nodes to source node $S_{1}$ is:(25)$$\begin{array}{l} P_{S_{2}S_{1}}=\frac{1}{2}P\left\{\varepsilon_{S_{2}S_{1}}^{S_{2h}}|\lambda ,\mu \right\}+\\ \frac{1}{2}\left[1-\left(1-P\left\{\varepsilon_{S_{4}R}^{S_{4h}}|\lambda \right\}\right)\left(1-P\left\{\varepsilon_{S_{2}R}^{S_{2l}}|\lambda \right\}\right)\left(1-P\left\{\varepsilon_{RS_{1}}^{J_{h}}|\lambda \right\}\right)\left(1-P\left\{\varepsilon_{S_{4}S_{1}}^{S_{2h}}|\lambda ,\mu \right\}\right)\right] \end{array}$$
(26)$$\begin{array}{l} P_{S_{4}S_{1}}=\frac{1}{2}P\left\{\varepsilon_{S_{4}S_{1}}^{S_{4h}}|\lambda ,\mu \right\}+\\ \frac{1}{2}\left[1-\left(1-P\left\{\varepsilon_{S_{2}R}^{S_{2h}}|\lambda \right\}\right)\left(1-P\left\{\varepsilon_{S_{4}R}^{S_{4l}}|\lambda \right\}\right)\left(1-P\left\{\varepsilon_{RS_{1}}^{J_{l}}|\lambda \right\}\right)\left(1-P\left\{\varepsilon_{S_{4}S_{1}}^{S_{4h}}|\lambda ,\mu \right\}\right)\right] \end{array}$$
(27)$$P_{S_{3}S_{1}}=1-\frac{1}{2}\left[\begin{array}{l} \left(1-P\left\{\varepsilon_{S_{1}R}^{S_{1h}}|\lambda \right\}\right)\left(1-P\left\{\varepsilon_{S_{3}R}^{S_{3l}}|\lambda \right\}\right)\left(1-P\left\{\varepsilon_{RS_{1}}^{J_{h}}|\lambda \right\}\right)+\\ \left(1-P\left\{\varepsilon_{S_{3}R}^{S_{3h}}|\lambda \right\}\right)\left(1-P\left\{\varepsilon_{S_{1}R}^{S_{1l}}|\lambda \right\}\right)\left(1-P\left\{\varepsilon_{RS_{1}}^{J_{l}}|\lambda \right\}\right) \end{array}\right]$$

The total bit error rate of source node $S_{1}$ can be expressed as:(28)$$P\left(\lambda ,\mu \right)=1-\left(1-P_{S_{2}S_{1}}\right)\left(1-P_{S_{3}S_{1}}\right)\left(1-P_{S_{4}S_{1}}\right)$$

### 4.2. System Channel Capacity Analysis

In a communication system, for a given signal-to-noise ratio (SNR) channel, according to Shannon’s theorem, the normalized channel capacity of the system can be expressed by the following Equation (29) [19]:(29)$$C/B=\log_{2}^{\left(1+SNR\right)}$$

In an FSO communication system, the relationship between the instantaneous signal-to-noise ratio of a channel SNR and the average signal-to-noise ratio of a channel is $SNR=\gamma_{0}h^{2}$. Affected by atmospheric turbulence, under the condition of fully correlated atmospheric fading, the channel capacity value fluctuates randomly, so the expected value of the channel capacity is used to describe the average channel capacity of the channel; its expression is Cerg=E(C(h)). According to a previous study [20], the average channel capacity of a channel can be expressed as:(30)$$Cerg=\int_{0}^{\infty }B\log_{2}\left(1+\gamma_{0}h^{2}\right)f\left(h\right)dh$$

In the formula, B is the channel bandwidth and $\gamma_{0}$ is the average signal-to-noise ratio of the channel. In the system shown in Figure 1, the signal-to-noise ratio of the source node to the relay node link, transmitting high priority data and low priority data, can be expressed as:(31)$$\gamma_{high}=\frac{{\left|h_{NR}\right|}^{2}\lambda P_{tr}}{\left(1-\lambda \right){\left|h_{NR}\right|}^{2}P_{tr}+N_{0}}$$
(32)$$\gamma_{low}=\frac{{\left|h_{NR}\right|}^{2}\left(1-\lambda \right)P_{tr}}{N_{0}}$$
where $h_{NR}$ is the channel status information of the source node to the relay node link, $N\in \left\{S_{1},S_{2},S_{3},S_{4}\right\}$. $N_{0}$ is the variance of the noise at the relay node R, and $P_{tr}$ is the signal transmission power. Similarly, the signal-to-noise ratio of sending high-priority data between adjacent source nodes can be expressed by Equation (33):(33)$$\gamma_{high}^{’}=\frac{{\left|h_{ij}\right|}^{2}\lambda P_{tr}}{\left(1-\lambda \right){\left|h_{ij}\right|}^{2}P_{tr}+N_{0}^{’}}$$
where $h_{ij}$ is the channel status information of the link between the adjacent source nodes, $i,j\in \left\{S_{1},S_{2},S_{3},S_{4}\right\}$. $N_{0}^{’}$ is the variance of the noise at source node j. Equation (1) is substituted (30), and the normalized average channel capacity of the channel can be obtained as:(34)$$Cerg/B=\frac{{\left(\alpha \beta \right)}^{\left(\alpha +\beta \right)/2}}{\Gamma \left(\alpha \right)\Gamma \left(\beta \right)}\int_{0}^{\infty }\log_{2}\left(1+\gamma_{0}h^{2}\right)h^{\left(\frac{\alpha +\beta }{2}\right)-1}G\begin{array}{c} 2,0\\ 0,2 \end{array}\left[\alpha \beta h|\begin{array}{c} -\\ \frac{\alpha -\beta }{2},-\frac{\alpha -\beta }{2} \end{array}\right]dh$$

Since the system operates in half-duplex mode, the normalized average channel capacity from the source node to the relay node can be expressed by the following equation:(35)$$Cerg_{NR}/B=\frac{{\left(\alpha \beta \right)}^{\left(\alpha +\beta \right)/2}}{\Gamma \left(\alpha \right)\Gamma \left(\beta \right)}\left(\begin{array}{l} \int_{0}^{\infty }\log_{2}\left(1+\gamma_{high}h_{NR}^{2}\right)h_{NR}^{\left(\frac{\alpha +\beta }{2}\right)-1}G\begin{array}{c} 2,0\\ 0,2 \end{array}\left[\alpha \beta h_{NR}|\begin{array}{c} -\\ \frac{\alpha -\beta }{2},-\frac{\alpha -\beta }{2} \end{array}\right]dh_{NR}\\ +\int_{0}^{\infty }\log_{2}\left(1+\gamma_{low}h_{NR}^{2}\right)h_{NR}^{\left(\frac{\alpha +\beta }{2}\right)-1}G\begin{array}{c} 2,0\\ 0,2 \end{array}\left[\alpha \beta h_{NR}|\begin{array}{c} -\\ \frac{\alpha -\beta }{2},-\frac{\alpha -\beta }{2} \end{array}\right]dh_{NR} \end{array}\right)$$

The normalized average channel capacity of the link between adjacent source nodes can be expressed by the following equation:(36)$$Cerg_{ij}/B=\frac{{\left(\alpha \beta \right)}^{\left(\alpha +\beta \right)/2}}{\Gamma \left(\alpha \right)\Gamma \left(\beta \right)}\int_{0}^{\infty }\log_{2}\left(1+\gamma_{high}^{’}h^{2}\right)h_{ij}{}^{\left(\frac{\alpha +\beta }{2}\right)-1}G\begin{array}{c} 2,0\\ 0,2 \end{array}\left[\alpha \beta h_{ij}|\begin{array}{c} -\\ \frac{\alpha -\beta }{2},-\frac{\alpha -\beta }{2} \end{array}\right]dh_{ij}$$

In the system shown in Figure 1, adjacent nodes are interconnected in pairs, and the total normalized average channel capacity of the system is:(37)$$Cerg_{total}/B=\sum_{i=1}^{M}\sum_{j=1}^{M}Cerg_{ij}/B+\sum_{i=1}^{M}Cerg_{N_{i}R}/B+2Cerg_{RN}/B$$
where M is the total number of source nodes.

## 5. Simulation Results and Analysis

In order to verify the bit error rate and average channel capacity performance of the FSO communication system shown in Figure 1, the atmospheric channels under different turbulence intensities were simulated, and the system simulations with different priority parameters and different channel asymmetric parameters were carried out. The simulation parameters are shown in Table 1:

Aiming at the impact of atmospheric turbulence with different intensity on the performance of the FSO communication system, that is, in this system, the stronger the turbulence intensity, the worse the BER performance at the user node and the total average channel capacity of the system. The source information was layer-modulated at the source node, and the received information was layer-modulated and network-encoded at the relay node to overcome the interference of atmospheric turbulence on the performance of the FSO communication system. In this study, because the transmission distance of each communication link was inconsistent, the intensity of the atmospheric turbulence channel is reflected by the atmospheric refractive index structure constant $C_{n}^{2}$, the source information is modulated by 4/16-QAM, and the received information is encoded by the physical-layer network at the relay node. By deducing the expressions of BER and the average channel capacity of the FSO communication system, the BER performance and average channel capacity performance of the system under different turbulence intensities, different constellation priority parameters, and different channel asymmetry parameters were analyzed.

In order to obtain the change of BER performance of the cooperative FSO communication system, with constellation priority parameters under different atmospheric turbulence intensity and different channel average signal-to-noise ratio, the system simulation, as shown in Figure 4, was carried out. In Figure 4, the atmospheric turbulence intensity is represented by the atmospheric refractive index structure constant $C_{n}^{2}$. We know that the bit error rate of the cooperative FSO communication system decreases with the increase of the constellation priority parameter. Under the same signal-to-noise ratio, the atmospheric refractive index structure constant $C_{n}^{2}$C increased. The stronger the atmospheric turbulence intensity, the greater the impact on the BER performance of the cooperative FSO communication system, so the BER performance of the system became worse and worse. Under the same atmospheric turbulence intensity, such as the atmospheric refractive index structure constant $C_{n}^{2}=1.2\times 10^{-14}$, the average signal-to-noise ratio of the channel increased from SNR=5 dB to SNR=15 dB, and the BER performance of the system became better and better, and could drop below $10^{-8}$. The higher the signal-to-noise ratio, the more obvious the BER performance of the system. The atmospheric refractive index structure constants $C_{n}^{2}=2.8\times 10^{-14}$ and $C_{n}^{2}=7.2\times 10^{-14}$ also had the same improvement effect. This shows that the layered modulation, combined with the physical-layer network coding scheme, has a better improvement effect on the high SNR channel.

Figure 5 shows the change of BER performance of the cooperative FSO communication system with channel asymmetry parameters. It can be seen from Figure 5 that under different atmospheric turbulence intensities and different channel average signal-to-noise ratios, the system bit error rate decreased with the increase of channel asymmetry parameters. Under the average signal-to-noise ratio of the same channel, the BER performance of the system decreased due to the enhancement of atmospheric turbulence intensity. Under the same atmospheric turbulence intensity, such as the atmospheric refractive index structure constant $C_{n}^{2}=1.2\times 10^{-14}$, the average signal-to-noise ratio of the channel increased from SNR=5dB to SNR=15dB, and the BER performance of the system became better and better, which could drop below $10^{-8}$. The higher the average signal-to-noise ratio of the channel, the more obvious the BER performance drops. When the atmospheric refractive index structure constants $C_{n}^{2}=2.8\times 10^{-14}$ and $C_{n}^{2}=7.2\times 10^{-14}$, the scheme also had the same improvement effect on the cooperative FSO communication system. To sum up, the higher the average signal-to-noise ratio of the channel, the more obvious the decrease of the bit error rate of the system using the layered modulation joint physical-layer network coding scheme, which indicates that the scheme has a better improvement effect on the channel with high signal-to-noise ratio.

In order to further obtain the improvement of the BER performance of the cooperative FSO communication system by using the layered modulation joint physical-layer network coding scheme under different atmospheric turbulence intensities, the simulation was carried out (Figure 6). In Figure 6, the atmospheric refractive index structure constants are $C_{n}^{2}=1.2\times 10^{-14}$, $C_{n}^{2}=2.8\times 10^{-14}$, and $C_{n}^{2}=7.2\times 10^{-14}$, respectively, and the simulation was carried out with and without the layered modulation-joint physical-layer network coding scheme. It can be seen from Figure 6 that the higher the atmospheric refractive index structure constant, the stronger the atmospheric turbulence intensity, and the lower the BER performance of the cooperative FSO communication system. Under different refractive index structure constants, the BER performance of cooperative FSO communication system can be improved by using layered modulation, combined with a physical-layer network coding scheme. When the BER of the system is $10^{-6}$ and the atmospheric refractive index structure constant $C_{n}^{2}=1.2\times 10^{-14}$, the BER performance of the system was improved by about 0.5 dB, by using the layered modulation-joint physical-layer network coding scheme; when the bit error rate of the system was $10^{-6}$ and the atmospheric refractive index structure constant $C_{n}^{2}=2.8\times 10^{-14}$, the bit error rate performance of the system using the layered modulation-joint physical layer network coding scheme was improved by about 1 dB; when the bit error rate of the system was $10^{-6}$ and the atmospheric refractive index structure constant $C_{n}^{2}=7.2\times 10^{-14}$, the bit error rate performance of the system using the layered modulation-joint physical-layer network coding scheme was improved by about 1.5 dB. In the strong atmospheric turbulence channel, the higher the BER performance of the cooperative FSO communication system using this scheme, the more it indicates that this scheme is applicable to the strong atmospheric turbulence channel and has higher anti-interference performance to the strong atmospheric turbulence channel.

At the same time, in order to further obtain the effect of hierarchical modulation combined with the physical-layer network coding scheme on the average channel capacity (C/B) of the system, the average channel capacity under different constellation priority parameters λ was studied in this paper. Figure 7 shows the relationship between the average channel capacity of the system and the constellation priority parameters under different signal transmission powers. The average channel capacity of the system increased with the increase of the constellation priority parameter, and the higher the signal transmission power, the higher the average channel capacity of the channel. This is because the increase in transmission power and constellation priority parameters ensures the signal-to-noise ratio of high priority data and improves the reliability of system data transmission, so the average channel capacity of the system increases.

Figure 8 shows the relationship between the system average channel capacity and constellation priority parameters under different atmospheric turbulence intensities. In Figure 8, the atmospheric turbulence intensity is reflected by the refractive index structure constant C1. The larger the refractive index structure constant, the stronger the atmospheric turbulence intensity. Under different atmospheric turbulence intensities, the average channel capacity of the system increased with the increase of constellation priority parameters. The strong atmospheric turbulence led to the deterioration of the channel conditions, which interfered with the growth of the average channel capacity of the system, so the average channel capacity performance of the strong atmospheric turbulence channel deteriorated. In conclusion, the layered modulation, combined with physical-layer network coding scheme, can improve the average channel capacity performance of the system, and the higher the constellation priority parameter, the better the average channel capacity performance of the system.

## 6. Conclusions

To solve the problem of system performance deterioration caused by channel asymmetry in the cooperative FSO communication system, this paper proposed a scheme of using layered modulation and joint physical-layer network coding in the cooperative FSO communication system. With the use of layered modulation for the transmitted source information, and using physical-layer network coding at the relay node, the performance of the cooperative FSO communication system is improved. Through the simulation and analysis of different constellation priority parameters and different channel asymmetry parameters, it can be concluded that the higher the constellation priority parameters and channel asymmetry parameters, the better the system performance. Under different atmospheric turbulence intensities, when the average signal-to-noise ratio of the channel rises from SNR=5 dB to SNR=15 dB, the bit error rate of the cooperative FSO communication system can be reduced to below $10^{-8}$, and under strong atmospheric turbulence channel conditions, the system can obtain a signal-to-noise ratio gain of about 1.5 dB. The higher the average signal-to-noise ratio of the channel, the faster the bit error rate of the system decreases. In atmospheric turbulence of different intensities, the average channel capacity of the system increases with the increase of the constellation priority parameters. The simulation results show that the layer modulation combined with the physical-layer network coding scheme can improve the BER performance and average channel capacity performance of the cooperative FSO communication system; it is suitable for the channel, with strong atmospheric turbulence and high signal-to-noise ratio, and shows better anti-interference performance for the channel with strong atmospheric turbulence.

## Figures and Tables

**Figure 1 sensors-22-06912-f001:**
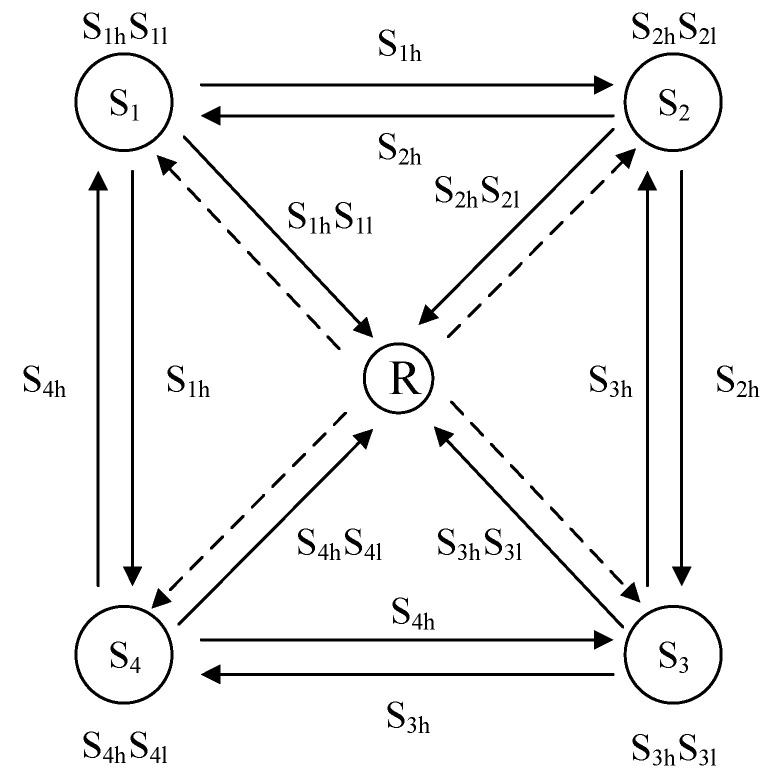
Relay network model with asymmetric channel conditions.

**Figure 2 sensors-22-06912-f002:**
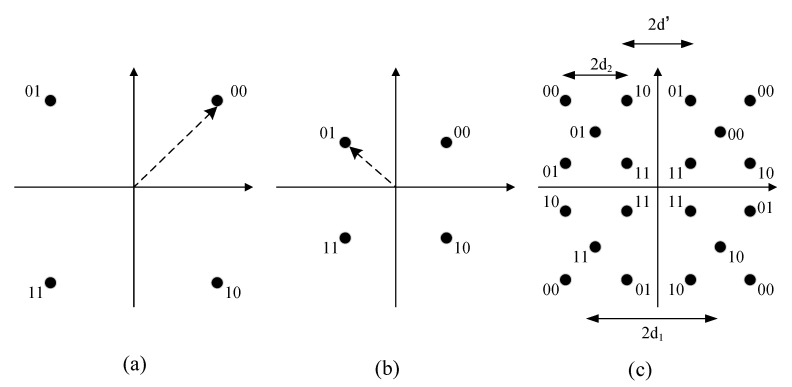
Constellation of 4/16-QAM hierarchical modulation.

**Figure 3 sensors-22-06912-f003:**
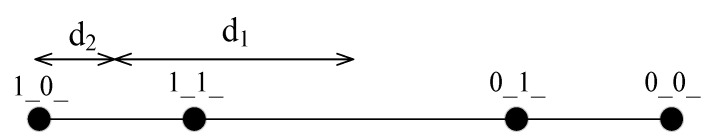
4/16-QAM modulated in-phase subflow.

**Figure 4 sensors-22-06912-f004:**
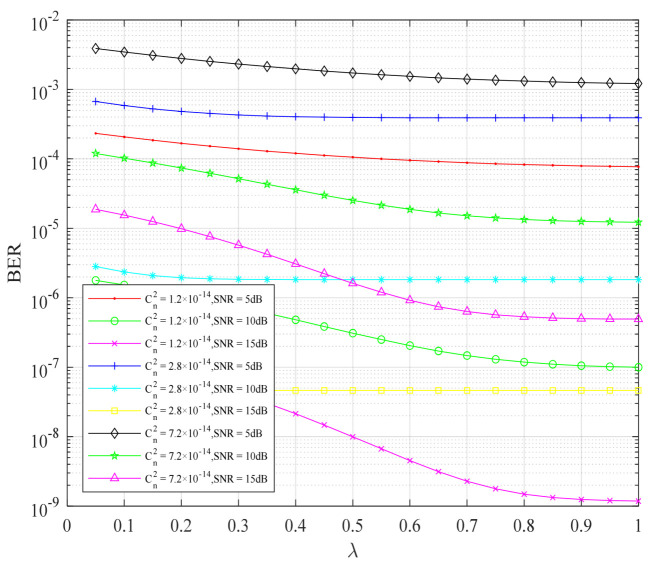
Comparison of system BER performance under different constellation priority parameters.

**Figure 5 sensors-22-06912-f005:**
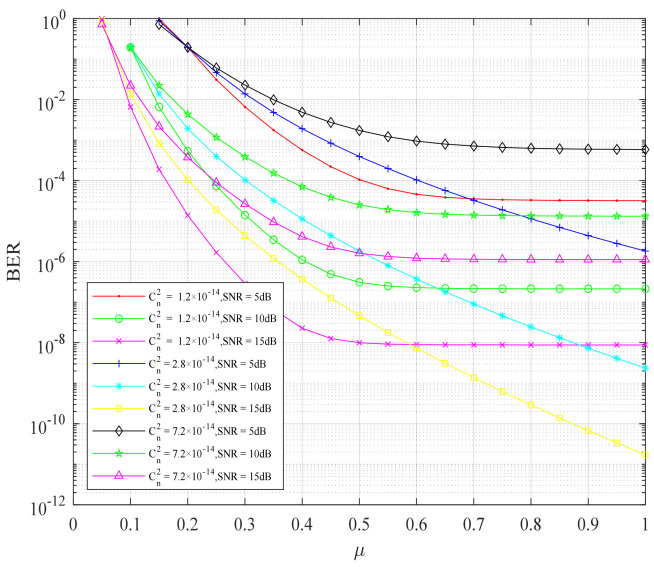
Comparison of system BER performance under different channel asymmetric parameters.

**Figure 6 sensors-22-06912-f006:**
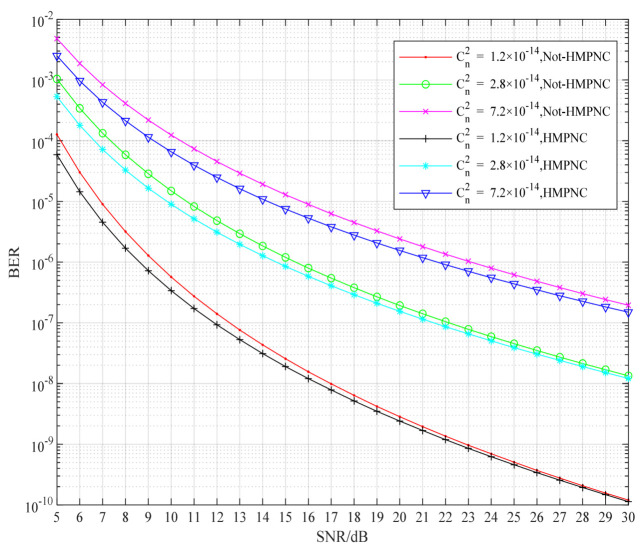
Comparison of the improvement of BER performance of the cooperative FSO communication system by this scheme under different turbulence intensities.

**Figure 7 sensors-22-06912-f007:**
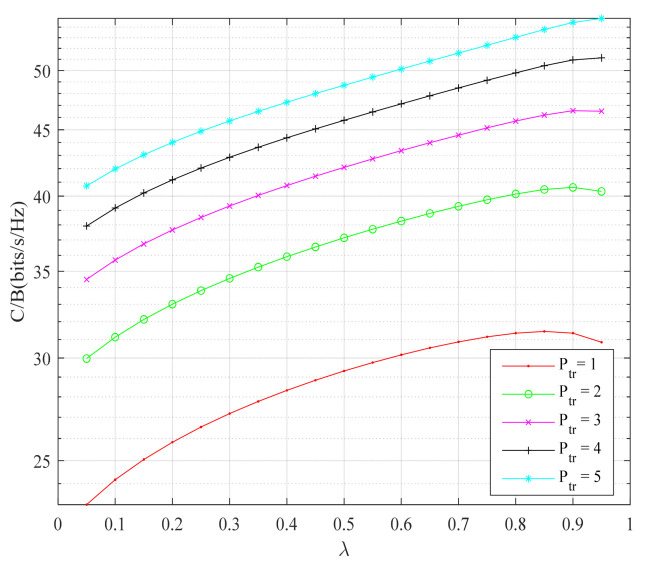
Average channel capacity of the system under different transmission powers and different constellation priority parameters.

**Figure 8 sensors-22-06912-f008:**
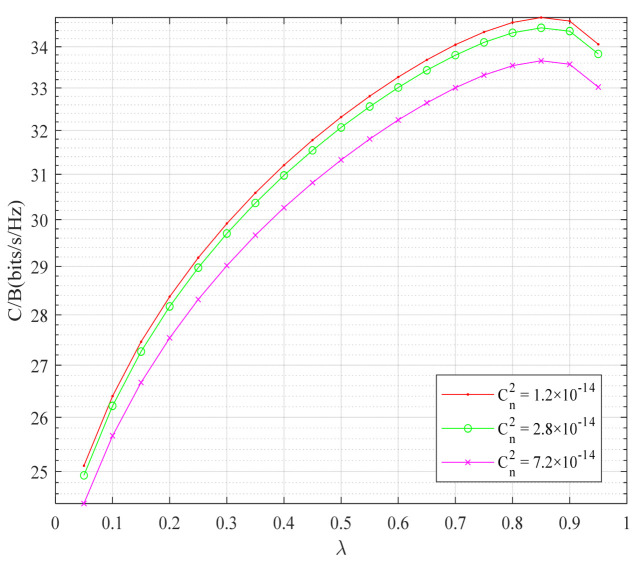
System average channel capacity under different turbulence intensities and constellation priority parameters.

**Table 1 sensors-22-06912-t001:** Simulation parameter settings.

Parameter Type	Value
Refractive index structure constant/(m^−2/3^)	1.2 × 10^−14^, 2.8 × 10^−14^, 7.2 × 10^−^^14^
Laser wavelength (m)	1.55 × 10^−6^
Hierarchical modulation mode Network code type	4/16-QAM Physical layer network code
Communication distance of direct link	5000 m
Communication distance of relay link	3535 m

## Data Availability

Not applicable.

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
