# Peer review of "Research on Performance of Cooperative FSO Communication System Based on Hierarchical Modulation and Physical Layer Network Code"

_sensors, 2022, doi:10.3390/s22186912_

Round 1
Reviewer 1 Report
Review of the manuscript Sensors-1909595
Research on performance of cooperative FSO communication system based on hierarchical modulation and physical layer network code
By
Humayun Qin, Yang Cao, Ibofanga Peng, Zupeng Zhang
The authors considered theoretically a hierarchical modulation joint physical layer network coding scheme in order to improve the collaborative communication system performance. They developed a mathematical model based on Gamma-Gamma (GG) distribution model. They carried out numerical simulations; they evaluated the bit error rate (BER) under different conditions including the atmospheric refractive indices; they presented the constellation of 4/16-QAM hierarchical modulation. The proposed manuscript contains novel results. It can be interesting for the researchers and engineers occupied in the field of communication systems.
The manuscript can be recommended for publication with the following minor revisions.
1. The authors used the abbreviation FSO without the explanation. Please, write down in the beginning of the manuscript what does this abbreviation mean.
2. Section 2. The system model is based on four source nodes . It is unclear whether this is a general case or a particular example. Were the models with other numbers of nodes considered? Is the number of nodes limited?
3. Page 4. Did the authors themselves derived equations (1), (2) and (3), or these relationships are known from the literature? In the former case, please explain the derivation; in the latter case, please give a corresponding reference.
4. Equations (6)-(10). The quantities, , are not defined in the text.
5. The bit error rate (BER) in equations (20)-(28) is defined as . But in equations (31)-(33) is defined as the signal transmission power (page 9, lines 284, 285).
6. Section 6. Conclusions. There is not any comparison of the simulation results obtained in the proposed manuscript with the experimental results.

Author Response
Response to Reviewer1 Comment
Point1: The authors used the abbreviation FSO without the explanation. Please, write down in the beginning of the manuscript what does this abbreviation mean.
Response 1: I have revised the 28th line of the paper. FSO is the abbreviation of free space optical.
Point2: Section 2. The system model is based on four source nodes S1,2,3,4 . It is unclear whether this is a general case or a particular example. Were the models with other numbers of nodes considered? Is the number of nodes limited?
Response 2: The system model composed of four nodes in the second section is a special structure, and this paper analyzes and discusses this special structure. This structure is representative to a certain extent and can be expanded to other structures and other node numbers. The number of nodes is not limited.
Point3: Page 4. Did the authors themselves derived equations (1), (2) and (3), or these relationships are known from the literature? In the former case, please explain the derivation; in the latter case, please give a corresponding reference.
Response 3 : equations (1), (2) and (3) are quoted from other references. I have marked the participating documents in the paper, and the reference number is [14].
Point4: Equations (6)-(10). The quantities , , and are not defined in the text.
Response 4 : is the codeword structure of the physical layer network coding used for the received information at the relay node, which has been marked in line 188 of the paper. is the mixed information formed by two network encoded codewords at the relay node, which has been marked in line 190. and are two variables in , which have been marked on line 193 of the paper.
Point5: The bit error rate (BER) in equations (20)-(28) is defined as P . But in equations (31)-(33)P is defined as the signal transmission power (page 9, lines 284, 285).
Response 5 : I have changed P in equations (31) - (33) to , and also changed P in Fig. 7 to .
Point6: Section 6. Conclusions. There is not any comparison of the simulation results obtained in the proposed manuscript with the experimental results.
Response 6: The conclusion part is concluded and summarized based on the simulation experiment results, and the data in the conclusion part is also the simulation experiment results. Through the simulation experimental data, it is concluded that the layered modulation joint physical layer network coding scheme can improve the performance of the cooperative FSO communication system, and has a better effect on the strong turbulent atmosphere environment.
I have modified other parts according to your suggestions. Please review them!

Reviewer 2 Report
This paper presented a layered modulation joint physical layer network coding scheme to improve the performance of asymmetric cooperative FSO communication systems. The simulation results were also presented and investigated to verify the effectiveness of the proposed scheme, but there are still some issues should be addressed before publication.
1. The characteristics of the hierarchical modulation combined with the network coding techniques in solving the asymmetric degradation for cooperative FSO communication should be addressed, especially when compared with existing techniques in the introduction part.
2. Since it seems the proposed scheme combined hierarchical modulation and network coding techniques, the novelty of the paper should be addressed.
3. Some mistakes in English should also be corrected, such as the sentence in Line 141.
4. In the simulation, the choice of the settings should be analyzed, such as Refractive index structure constant, and the Hierarchical modulation mode.
Author Response
Response to Reviewer2 Comment
Point1: The characteristics of the hierarchical modulation combined with the network coding techniques in solving the asymmetric degradation for cooperative FSO communication should be addressed, especially when compared with existing techniques in the introduction part.
Response 1: Aiming at the asymmetric reception problem of cooperative FSO communication system, this paper proposes to use layered modulation combined with physical layer network coding technology to solve this problem. In the third part of this paper, we point out how to combine layered modulation and physical layer network coding in cooperative FSO communication system.
Point2: Since it seems the proposed scheme combined hierarchical modulation and network coding techniques, the novelty of the paper should be addressed.
Response 2: According to my recent research, there are not many scholars who have conducted in-depth research on the asymmetric reception of cooperative FSO communication systems. Therefore, I think the novelty of this paper lies in the use of layered modulation combined with physical layer network coding to solve the asymmetric reception problem of cooperative FSO communication systems. This has been pointed out in the summary. If you think the content about novelty is not enough, I will add it in the subsequent revision. thank you!
Point3: Some mistakes in English should also be corrected, such as the sentence in Line 141.
Response 3 : Thank you for pointing out the error. I have corrected it.
Point4: In the simulation, the choice of the settings should be analyzed, such as Refractive index structure constant, and the Hierarchical modulation mode.
Response 4: The setting of simulation parameters are some basic parameters for building FSO communication links. The atmospheric refractive index structure constant is set to characterize the turbulence intensity of different intensities, which has been pointed out in the paper. The layered modulation mode is 4 / 16-QAM modulation, which is the most suitable modulation mode for FSO communication system at present.
I have modified other parts according to your suggestions. Please review them!

Reviewer 3 Report
The author proposed Research on Performance of Cooperative FSO Communication 2 System Based on Hierarchical Modulation and Physical Layer 3 Network Code.
In the strong atmospheric turbulence channel, the cooperative FSO com- 20 munication system can obtain a signal-to-noise ratio gain of about 1.5dB,How does it obtained and any previous study claim the gain above this?
Highlights and contribution of the paper may be added
In figure terminologes like s1h --s4h need to be mentioned properly
The results and technical experimentation are neatly explained
please read the survey on Comparative analysis of FSO, OFC and diffused channel links in Photonics using Artificial Intelligence based S-BAND,C-BAND and L-BAND of the attenuation metrics.
Author Response
Response to Reviewer3 Comment
Point1: In the strong atmospheric turbulence channel, the cooperative FSO com- 20 munication system can obtain a signal-to-noise ratio gain of about 1.5dB,How does it obtained and any previous study claim the gain above this?
Response1: The signal-to-noise ratio gain is obtained according to Fig. 6, which compares the use of layered modulation and joint physical layer network coding with the use of no such scheme. The data of 1.5dB is obtained when the system bit error rate is 10-6 under strong turbulence.
Point2:Highlights and contribution of the paper may be added
Response2: The key points and contributions of this article have been pointed out in the abstract and introduction. If it needs to be further pointed out in detail, I will add content in subsequent revisions.
Point3:In figure terminologes like s1h --s4h need to be mentioned properly
Response3: is the high priority data sent by each node, as indicated in line 169 of the paper.
Point4:The results and technical experimentation are neatly explaine
Response4: Thank you!
Point5:please read the survey on Comparative analysis of FSO, OFC and diffused channel links in Photonics using Artificial Intelligence based S-BAND,C-BAND and L-BAND of the attenuation metrics.
Response5: Thank you for your suggestions. I will take your suggestions and read the relevant content. The communication system model built in this paper has certain universality, and the related technology has been applied in Chengdu. If necessary, I will add this part in the subsequent modification according to your suggestion.
I have modified other parts according to your suggestions. Please review them!
